# Occurrence and Removal of Priority Substances and Contaminants of Emerging Concern at the WWTP of Benidorm (Spain)

**Edmond Tiberius Alexa [1,\*], María de los Ángeles Bernal-Romero del Hombre Bueno [1], Raquel González [2], Antonio V. Sánchez [3], Héctor García [4] and Daniel Prats [1]**

[1] University Institute of the Water and the Environmental Sciences (IUACA), Universidad de Alicante, 03690 San Vicente del Raspeig, Spain

[2] Laboratory of Biotechnology and Synthetic Biology, Institute for Integrative Systems Biology University of Valencia, 46980 Paterna, Spain

[3] Laboratorios Tecnológicos de Levante SL, 46980 Paterna, Spain

[4] Water Supply, Sanitation and Environmental Engineering Department, IHE Delft Institute for Water Education, 2611 AX Delft, The Netherlands

\* Correspondence: eta10@alu.ua.es

**Abstract:** This work is part of the European research project LIFE15 ENV/ES/00598 whose objective was to develop an efficient and sustainable methodology to eliminate Priority Substances (PS) and Contaminants of Emerging Concern (CEC), in Wastewater Treatment Plants (WWTP). The aim was to achieve reduce the concentration of PSs until their concentration was below the quality limit established in the DIRECTIVE 2013/39/EU, and to achieve reductions of 99% of the initial concentration for the selected CECs. The plant selected for the experimentation was the Benidorm WWTP (Spain). This publication studied the appearance and elimination, in the conventional treatment of this plant, of 12 priority substances (EU) and 16 emerging pollutants (5 of them included in the EU watch lists) during a year of experimentation. The analytical methods of choice were High Performance Liquid Chromatography coupled to a Mass Spectrometer (HPLC-MS/MS) and Gas Chromatography coupled to a Mass Spectrometer (GC-MS/MS). Results showed that the PSs atrazine, brominated diphenyl ether, isoproturon, octylphenol, pentachlorobenzene, simazine, terbutryn, tributyltin, and trifluralin, and the CECs 17-$\alpha$-ethinylestradiol, 17-$\beta$-estradiol, imazalil, orthophenylphenol, tertbutylazine, and thiabendazole, were not detected. The micropollutants with the highest average percentages of removal (>90%) are: chloramphenicol (100%), estriol (100%) and ibuprofen (99%). Partially removed were ketoprofen (79%), chlorpyrifos (78%), di(2-ethylhexyl) phthalate (78%), estrone (76%), sulfamethoxazole (68%), and fluoxetine (53%). The compounds with the lowest average percentage of removal (<50%) are diclofenac (30%), erythromycin (1%), diuron (0%) and carbamazepine (0%). For the micropollutants chlorpyrifos, diclofenac, erythromycin, sulfamethoxazole, carbamazepine, fluoxetine, ibuprofen, and ketoprofen, complementary treatments will be necessary in case there is a need to reduce their concentrations in the WWTP effluent below a certain standard. The presence of the different micropollutants in the samples was not regular. Some of them were presented continuously, such as carbamazepine; however, others sporadically such as chloramphenicol and others were associated with seasonal variations or related to remarkable periods of time, such as sulfamethoxazole.

**Keywords:** wastewater treatment; priority substances; contaminants of emerging concern

## 1. Introduction

Due to the steady increase in human population, one of the biggest challenges in current times is the availability of appropriate quality water for human consumption and



other uses. One of the important aspects related to quality is the presence of priority and emerging pollutants in waters, due to the associated health and environmental risks [1,2]. Contaminants enter the water through various routes, whether through the sewage network, by direct discharge into surface water by hospitals, homes, or industrial plants, or directly through the soil of agricultural land treated with pesticides [1–5]. Even though the concentration of the pollutants can be low, between micrograms to nanograms per litre, that does not mean that they are excluded from having an adverse impact on the aquatic ecosystem and on human health [2,5,6].

A key question is how many potential pollutants we have to control to guarantee the quality of the waters. The number of chemicals used industrially is enormous. In a recent study by Wang et al. [7], after analysing 22 chemical inventories from 19 countries and regions to find a complete overview of chemicals on the market, more than 350,000 chemicals and chemical mixtures are identified, presenting up to three times more than previously estimated. The characteristics of many of these chemicals are publicly unknown because they are declared confidential (over 50,000) or ambiguously described (up to 70,000). Dulio et al. [8] report that between 30,000 and 50,000 industrial chemical substances are contained in "everyday products". Therefore, it is currently impossible to monitor all the chemicals present in water that may be potentially dangerous. The European Union, EU, has identified 45 priority substances, PS, for which it has established environmental quality standards [9]. On the other hand, in the EU, watch lists of contaminants of emerging concern, CEC, are established periodically for the purpose of supporting future prioritization of the selected substances. Three watch lists have already been established [10–12].

Wastewater is one of the main sources of priority and emerging pollutants in water bodies [13–18], so it is very important to ensure proper treatment of wastewater. The removal efficiency is highly affected by the physicochemical properties of the substances, type of treatment, and operating conditions [4,19–23]. For example, for many hydrophilic organic micropollutants, biological transformation will be the main disposal mechanism during wastewater treatment. The fraction of contaminant removed by biodegradation in the secondary treatment depends mainly on the amount of microorganisms present (i.e., the concentration of the activated sludge), the type of microorganisms (composition of the activated sludge), the biodegradability of the contaminant by these microorganisms (determinated by the biodegradation constant, kbiol), and the hydraulic retention time inside the reactor (since degradation usually follows a pseudo first order kinetics). The rate of biodegradation can also be influenced by temperature, (the higher the temperature the higher the reaction rate) pH (influences enzyme activity and cell absorption, with generally higher absorption of neutral species (no charge), redox conditions (usually higher under aerobic conditions) and availability of a co-substrate [24].

This work was part of the European research project LIFE15 ENV/ES/00598 whose objective was to develop an efficient and sustainable methodology to eliminate PS and CEC in WWTPs. The aim was to achieve reductions in PS until their concentration was below the quality limit established in the DIRECTIVE 2013/39/EU, and to achieve reductions of 99% of the initial concentration for the selected CECs. The selected plant was the Benidorm WWTP (Spain). In this publication, the occurrence and removal of 12 priority sub-stances (EU) and 16 contaminants of emerging concern (5 of them included in the EU watch lists) are presented. The results of this research provided valuable information on the behavior of the selected micropollutants and served as the basis for the design of the processes included in the pilot plant for the treatment of purified water in the WWTP. In future publications, the results of our research of the pilot plant for water purification treatment will be presented and compared to existing tertiary treatments of the WWTP of Benidorm.

## 2. Materials and Methods

### 2.1. Substances studied

Benidorm is a tourist city, with a high occupancy throughout the year, and with a significant presence of older people. Consequently, the presence of pharmaceuticals in its wastewater is foreseeable. In addition, in the municipal term agricultural activities are developed and there are some service industries. For this reason, substances that could originate from these activities have been selected, some of them priority substances and other contaminants of emerging concern. Twenty-eight substances were selected (twelve PS and sixteen CEC) that are shown in Table 1.

**Table 1.** Selected substances.

| Substance | Origin | Classification | | | |
|---|---|---|---|---|---|
| | | PS | CEC of the First Observation List (Commission Implementing Decision (EU) 2015/495) | CEC of the Third Observation List (Commission Implementing Decision (EU) 2020/1161) | Other CECs of Interest |
| 17-$\alpha$-ethinylestradiol | Pharm | | X | | |
| 17-$\beta$-estradiol | Pharm | | X | | |
| Atrazine | Ind/Agr | X | | | |
| Brominated diphenyl ether | Ind/Agr | X | | | |
| Carbamazepine | Pharm | | | | X |
| Chloramphenicol | Pharm | | | | X |
| Chlorpyrifos | Ind/Agr | X | | | |
| Di(2-ethylhexy)l phthalate | Ind Agr | X | | | |
| Diclofenac | Pharm | | X | | |
| Diuron | Ind/Agr | X | | | |
| Erythromycin | Pharm | | X | | |
| Estriol | Pharm | | | | X |
| Estrone | Pharm | | | | X |
| Fluoxetine | Pharm | | | | X |
| Ibuprofen | Pharm | | | | X |
| Imazalil | Ind/Agr | | | | X |
| Isoproturon | Ind/Agr | X | | | |
| Ketoprofen | Pharm | | | | X |
| Octylphenol | Ind/Agr | X | | | |
| Orthophenylphenol | Ind/Agr | | | | **X** |
| Pentachlorobenzene | Ind/Agr | X | | | |
| Simazine | Ind/Agr | X | | | |
| Sulfamethoxazole | Pharm | | | X | |
| Terbuthylazine | Ind/Agr | | | | X |
| Terbutryn | Ind/Agr | X | | | |
| Thiabendazole | Ind/Agr | | | | X |
| Tributyltin | Ind/Agr | X | | | |
| Trifluralin | Ind/Agr | X | | | |

*2.2. Sampling*

The water sample collection took place in the conventional WWTP of Benidorm. This plant is located in the province of Alicante, Spain. In 2021, it treated an average flow of 28,900 m³/day, corresponding to a served population of 200,500 equivalent inhabitants. The wastewater treatment plant is based on a conventional activated sludge process, CAS, of medium load, as follows: Influent reaches the WWTP through the headworks consisting of a bar screen, grit and fats removal, and primary clarifier. Then, the primary effluent reaches the secondary treatment, consisting of a CAS process with an anoxic basin for denitrification, following an aerobic basin for carbon removal and nitrification; there is intern recirculation from the aerobic basin to the anoxic basin. The sludge coming from aerobic basin is settled in a secondary clarifier; from then, some sludge is recirculated back to the aerobic basin, and the fraction remaining is wasted to set the solid retention time. The tertiary treatment uses ultrafiltration and reverse osmosis, finishing off the wastewater treatment by disinfecting it by the process of chlorination [25]. The key operational conditions of the secondary treatment are as follows: (i) hydraulic retention time of 13–15 h, (ii) dissolved oxygen concentration of 1,7 mg/L in the aerobic reactor; (iii) mixed liquor suspended solids concentration of 2,6 g/L; and (iv) solid retention time 8–10 days. Other parameters such as the temperature and presence and concentration of the conventional contaminants and emerging contaminants in the influent wastewater do fluctuate depending on the season.

The sampling period was 1 year, with the 1st sample collected on 24 November 2016. Two sampling points were selected. The first sampling point was after pre-treatment, from now on we will refer to this first sampling point as "influent". The CAS outlet was chosen as the second sampling point, henceforth we will refer to this second sampling point as "effluent" (Figure 1).

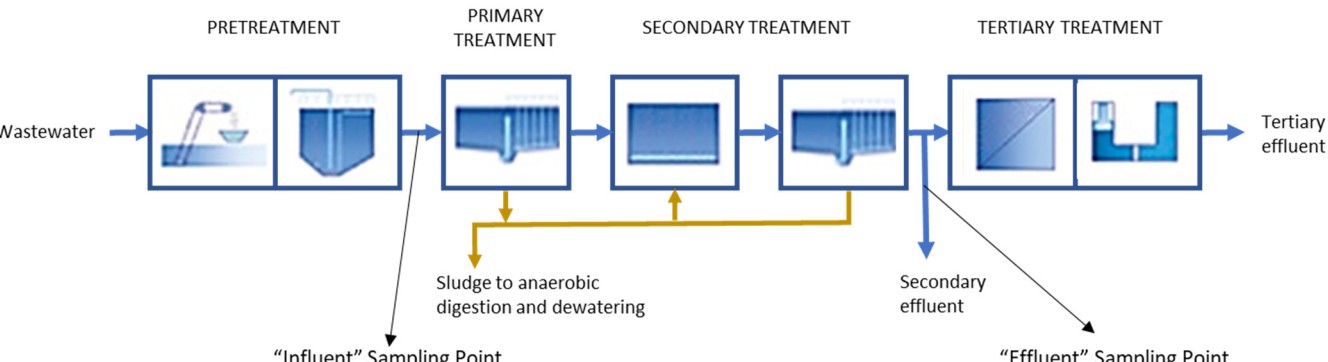

**Figure 1.** Sampling points in the water line of the Benidorm WWTP (process images adapted from EPSAR [24]).

One weekly sample of the influent and effluent was collected, up to a total of 108 samples (54 samples for each of the points). Two automatic samplers were used, one for the influent samples and the other for the effluent samples, which took 1 sample every hour, that is, the final sample analyzed corresponded to an integrated 24-h sample.

*2.3. Analytical Methodology*

For this study, the analytical methods selected were gas chromatography coupled to mass spectrometry (GC-MS/MS) and high-pressure liquid chromatography coupled to mass spectrometry (HPLC-MS/MS). Due to the very low concentrations of analytes in the water samples, a concentration step was necessary. For this, a Sorptive Stir Bar Extraction (SBSE) and a Solid Phase Extraction (SPE) were used in order to extract and pre-concentrate the analytes. Both concentration procedures were applied online just before analysis. The samples were analyzed by Laboratorios Tecnológicos de Levante S.L.

### 2.3.1. GC-MS/MS

An Agilent 7890 gas chromatograph coupled to an Agilent 7000 Triple Quad GC/MS mass spectrometer was used. For the process of pre-concentrating the samples, a Gerstel SBSE Twister system integrated in the instrument and an HP-5MS capillary column 30 m × 0.25 mm × 0.25 μm were used. To minimize instrumental error and possible matrix effects, the internal standard method was used. The samples were volatized and injected into the chromatographic column. In order for the analytes to elute, an inert gas was used as the mobile phase to not interact with the samples molecules and different temperatures are applied to the column.

### 2.3.2. HPLC-MS/MS

An Agilent 6460 Liquid Chromatograph coupled to a Triple Quad LC/MS Mass Spectrometer with a 3.0 mm × 120 mm × 2.7 μm Phenylhexyl Poroshell 120 column was used. The sample was pre-concentrated with an Autotrace 280 system on-line Thermo SPE integrated into the instrument. During the preparation of the samples, the Internal Standard method was followed to eliminate any possible instrumental error and the matrix effect caused by the analyzed samples. This technique allowed us to separate ionic species, macromolecules, labile products, polymeric materials, and a wide variety of high molecular weight compounds. One of the main advantages of this method is that it does not limit the samples due to its thermal stability and volatility.

Table 2 shows the analytical technique used for each compound, the concentration method, and the limit of quantification (LOQ). The LOQ corresponds to the concentration of the limit of detection multiplied by 3,3. In some cases, chromatogram baseline disturbances lead to relatively high LOQ values. In addition, the presence of organic matter and other contaminants in real wastewaters hinders the detection of trace contaminants.

**Table 2.** Analytical technique used for each compound.

| Analyte | Analytical Technique | Extraction and/or Preconcentration Method | LOQ μg L$^{-1}$ |
|---|---|---|---|
| 17-$\alpha$-ethinylestradiol | HPLC-MS | SPE | 0.05 |
| 17-β-estradiol | HPLC-MS | SPE | 0.005 |
| Atrazine | HPLC-MS | SPE | 0.05 |
| Brominated diphenyl ether | GC-MS | SBSE | 0.1 |
| Carbamazepine | HPLC-MS | SPE | 0.05 |
| Chloramphenicol | HPLC-MS | SPE | 0.005 |
| Chlorpyrifos | GC-MS | SBSE | 0.03 |
| Di(2-ethylhexyl) phthalate | GC-MS | SBSE | 1 |
| Diclofenac | HPLC-MS | SPE | 0.01 |
| Diuron | HPLC-MS | SPE | 0.05 |
| Erythromycin | HPLC-MS | SPE | 0.05 |
| Estriol | HPLC-MS | SPE | 0.1 |
| Estrone | HPLC-MS | SPE | 0.005 |
| Fluoxetine | HPLC-MS | SPE | 0.05 |
| Ibuprofen | HPLC-MS | SPE | 0.05 |
| Imazalil | HPLC-MS | SPE | 0.5 |
| Isoproturon | HPLC-MS | SPE | 0.05 |
| Ketoprofen | HPLC-MS | SPE | 0.05 |
| Octylphenol | GC-MS | SBSE | 0.03 |
| Orthophenylphenol | GC-MS | SBSE | 0.03 |
| Pentachlorobenzene | GC-MS | SBSE | 0.003 |
| Simazine | HPLC-MS | SPE | 0.05 |

| Sulfamethoxazole | HPLC-MS | SPE | 0.05 |
| Terbuthylazine | HPLC-MS | SPE | 0.05 |
| Terbutryn | GC-MS | SBSE | 0.05 |
| Thiabendazole | HPLC-MS | SPE | 0.5 |
| Tributyltin | GC-MS | SBSE | 0.0001 |
| Trifluralin | GC-MS | SBSE | 0.03 |

### 3. Results and Discussion

Table 3 shows mean influent and effluent concentrations (annual average), overall removal, and established environmental quality standards (EU) for priority substances.

**Table 3.** Mean influent and effluent concentrations (annual average), mean global removal, and established environmental quality standards (EU).

| Substance | Influent Average (µg L$^{-1}$ ± Standard Deviation) | Effluent Average (µg L$^{-1}$ ± Standard Deviation) | Overall Removal (%) | Established Environmental Quality Standards (EU) (µg L$^{-1}$) |
|---|---|---|---|---|
| 17-$\alpha$-ethinylestradiol * | <LOQ | <LOQ | - | |
| 17-β-estradiol | <LOQ | <LOQ | - | |
| Atrazine ** | <LOQ | <LOQ | - | 0.6 |
| Brominated diphenyl ether | <LOQ | <LOQ | - | - |
| Carbamazepine | 0.359 ± 0.081 | 0.360 ± 0.081 | 0 | |
| Chloramphenicol | 0.006 ± 0.010 | <LOQ | 100 | |
| Chlorpyrifos | 0.235 ± 0.067 | 0.051 ± 0.031 | 78 | 0.03 |
| Di(2-ethylhexyl) phthalate | 4.49 ± 0.291 | 0.972 ± 0.135 | 78 | 1.3 |
| Diclofenac | 1.61 ± 0.174 | 1.13 ± 0.146 | 30 | |
| Diuron | 0.048 ± 0.030 | 0.062 ± 0.034 | 0 | 0.2 |
| Erythromycin | 0.196 ± 0.061 | 0.195 ± 0.061 | 1 | |
| Estriol | 0.639 ± 0.108 | <LOQ | 100 | |
| Estrone | 0.038 ± 0.026 | 0.009 ± 0.013 | 76 | |
| Fluoxetine | 0.155 ± 0.053 | 0.073 ± 0.036 | 53 | |
| Ibuprofen | 31.8 ± 0.760 | 0.397 ± 0.026 | 99 | |
| Imazalil | <LOQ | <LOQ | - | - |
| Isoproturon | <LOQ | <LOQ | - | 0.3 |
| Ketoprofen | 2.54 ± 0.215 | 0.534 ± 0.098 | 79 | |
| Octylphenol | <LOQ | <LOQ | - | 0.1 |
| Orthophenylphenol | <LOQ | <LOQ | - | - |
| Pentachlorobenzene | <LOQ | <LOQ | - | 0.007 |
| Simazine | <LOQ | <LOQ | - | 1 |
| Sulfamethoxazole *** | 0.699 ± 0.113 | 0.226 ± 0.064 | 68 | |
| Terbuthylazine | <LOQ | <LOQ | - | - |
| Terbutryn | <LOQ | <LOQ | - | 0.065 |
| Thiabendazole | <LOQ | <LOQ | - | - |

| Tributyltin | <LOQ | <LOQ | - | 0.0002 |
| Trifluralin | <LOQ | <LOQ | - | 0.03 |

(*) Cells with pink background refer to CEC of the first observation list (EU). (**) Cells with a yellow background refer to priority substances (EU). (***) Cell with green background refer to CEC of the third observation list (EU).

The table reflects mean concentrations of 54 integrated influent samples and another 54 integrated effluent samples. The following results can be noted:

The PSs atrazine, brominated diphenyl ether, isoproturon, octylphenol, pentachlorobenzene, simazine, terbutryn, tributyltin, and trifluralin were not detected.

The PS **di(2-ethylhexyl) phthalate (DEHP)**, was present in the influent in a significant average concentration, 4.49 µg L$^{-1}$, and 78% was removed until reaching a concentration in the effluent below the established environmental quality standards.

The PS **chlorpyrifos** was present in the influent at an average concentration of 0.235 µg L$^{-1}$, and 78% is removed, but the average concentration in the effluent exceeded the established environmental quality standards for inland surface waters [9]. Therefore, to reduce the concentration of this PS to environmentally acceptable values, it is necessary to apply complementary treatments to the effluent of the biological process.

The PS **diuron** was present in the influent and effluent at very low average concentrations, 0.048 µg L$^{-1}$ and 0.062 µg L$^{-1}$ respectively, very close to the LOQ. The removal percentage would be negative, probably due to the very low values detected. In any case, the average concentration in the effluent was lower than the established environmental quality standards.

The CECs **17-$\alpha$-ethinylestradiol** and **17-$\beta$-estradiol**, which are included in the first EU observation list (10), were not detected.

The **CEC diclofenac** is an anti-inflammatory pharmaceutical included in the first EU observation list (10), which was detected in the influent in average concentrations of 1.61 µg L$^{-1}$. This concentration was in the high range of concentrations reported in wastewater (4, 18), which is reasonable given the high presence of elderly population in Benidorm. The average percentage of removal was very low, 30%, which was consistent with the majority of studies reported in the literature on biological treatments. [4,18,26]. To reduce the concentration of this CEC it is necessary to apply complementary treatments to the effluent of the biological process.

The CEC **erythromycin** is an antibiotic macrolide also included in the first EU observation list [10], which was detected in the influent in average concentrations of 0.196 µg L$^{-1}$. This concentration was in the higher range than other concentrations reported in wastewater [4,26,27], which is explained by the high presence of elderly people in Benidorm. The mean removal percentage was negligible, 1%, which indicated that it was not degraded or retained in sludge. This result was consistent with the one reported by Pasquini et al. [28] who did not find elimination of erythromycin in the liquid phase and verified that this antibiotic was also not adsorbed on the particulate matter or the sludge. On the other hand, other authors found removal of 42.8% in biological treatment [29]. To reduce the concentration of this CEC it is necessary to apply complementary treatments to the effluent of the biological process.

**Sulfamethoxazole** is a pharmaceutical antibiotic included in the third EU observation list [12]. Its average concentration in the influent was 0.699 µg L$^{-1}$. Very wide ranges of occurrence were reported in the literature, with values of 0.1–2.9 µg L$^{-1}$ in the raw influent to municipal WWTPs [18,26,27,29], being the most frequently detected antibiotic in all regions of the world, particularly in Asia [21]. The average elimination found was 68%. The removal of sulfamethoxazole reported in the literature had a very wide range, from negative values to 100% [18,21,26,27,29]. To reduce the concentration of this CEC it is necessary to apply complementary treatments to the effluent of the biological process.

The CECs **imazalil** (fungicide), **orthophenylphenol** (biocide), **terbutylazine** (herbicide), and **thiabendazole** (fungicide) were not detected.

The CECs carbamazepine (antiepileptic), chloramphenicol (antibiotic), estriol (estrogen), estrone (estrogen), fluoxetine (pharmaceutical antidepressant), ibuprofen (nonsteroidal anti-inflammatory drug), and ketoprofen (nonsteroidal anti-inflammatory drug) were detected:

**Carbamazepine** is an antiepileptic drug used to control seizures and is one of the most frequently detected pharmaceutical residues in bodies of water. It was even proposed as a substance to detect anthropogenic contamination in bodies of water [30]. The mean concentration in the influent was 0.359 µg L$^{-1}$, which fell within a range from LOQ to 18,500 ng L$^{-1}$ reported in a review work for full-scale WWTPs [26,27,31]. The average removal found was none, so carbamazepine could be considered a contaminant that is refractory to conventional biological treatment. Similar results were found in the literature [30–33], although removals of up to 94.9% were reported [26]. To reduce the concentration of this CEC it is necessary to apply complementary treatments to the effluent of the biological process.

**Chloramphenicol** is a broad-spectrum bacterial antibiotic. The average concentration in the influent was 0.006 µg L$^{-1}$, which is very close to the LOQ, which indicated a presence within a range similar to that found in other treatment plants in Europe [31]. Other reviews reported mean concentrations in the influent of 1.0 µg L$^{-1}$ [27]. The average removal found was very high, 100%, although this value could be imprecise since the effluent concentrations were detected below the LOQ. In the literature, removal percentages of 11.8–73.8 [34] and 76% [35] were indicated.

**Estriol** is an estrogen with mean concentration in the influent of 0.639 µg L$^{-1}$, that was in the high range of those reported in the literature [26,35,36]. The removal percentage for estriol was very high, 100%. This high percentage of elimination agreed with most of those reported in other WWTPs [35,36].

**Estrone** is an estrogen with mean concentrations in the influent of 0.038 µg L$^{-1}$, that was, in the middle range of what was reported in the literature [35,36]. The percentage of elimination of estrone could be considered high, 76%. This elimination percentage agreed with data from other WWTPs [35,36]. For this pharmaceutical CEC we considered that it was necessary to apply complementary treatments to the effluent of the biological process

**Fluoxetine** is a pharmaceutical antidepressant that is globally present in wastewater. In our influent, average concentrations of 0.155 µg L$^{-1}$ wer recorded, which were within the range of 0.018–2 µg/L$^{-1}$ found in the literature [26,29,36]. The average global removal percentage was 53%, which is also between the reported values 33 a 95% [26,29,37]. To reduce the concentration of this CEC it is necessary to apply complementary treatments to the effluent of the biological process.

**Ibuprofen** is a widely used nonsteroidal anti-inflammatory drug (NSAID). In our study, it was the micropollutant with the highest average concentration in the influent, 31.8 µg L$^{-1}$. This concentration confirmed other previous studies such as the one carried out by Santos et al. [38] who found ibuprofen was the most abundant compound detected in the influent of four WWTPs in Spain, with the concentration levels ranging from 3.73 to 603 µg L$^{-1}$. The average removal found was very high, 99%, although the average concentration in the effluent was still significant, 0.397 µg L$^{-1}$, due to the high concentration in the influent.

**Ketoprofen** is another NSAID that was quantified in influent water at an average concentration of 2.54 µg L$^{-1}$. This concentration could be considered in the highest range of those previously reported in the consulted literature [18,21,27,29,38,39]. The average global removal was 79%. The range of removal efficiency reported in the literature went from 0 to 100% [18,21,27,29,38,39]. To reduce the concentration of this CEC it is necessary to apply complementary treatments to the effluent of the biological process.

As a summary of the results obtained in the year of monitoring and measurement of 12 priority substances 16 pollutants of emerging concern in the primary and secondary treatment of the Benidorm treatment plant, it should be noted:

Of the 12 PSs studied, 9 of them registered average concentrations below the LOQ and 1 close to the LOQ. The DEHP was present in the influent in a significant average concentration, but was removed until a concentration in the effluent was below the established environmental quality standards. The chlorpyrifos was present in the influent at a significant average concentration, and the average concentration in the effluent exceeded the established environmental quality standards. Therefore, it is necessary to apply complementary treatments to the effluent of the biological process for meeting the standards.

Of the 16 CECs studied, 5 of them registered average concentrations below the LOQ and 1 close to the LOQ. For another 3, very low concentrations were found in the effluent. For the rest of the CECs diclofenac (first observation list), erythromycin (first observation list), sulfamethoxazole (third observation list), carbamazepine, estrone, fluoxetine, ibuprofen, and ketoprofen, additional treatments to the effluent of the biological process should be applied. In reality, tertiary treatment is applied in the WWTP (Figure 1).

The presence of the different micropollutants in the samples was not regular. Some of them were observed continuously; however, others sporadically, and others were associated with seasonal variations or related to remarkable periods of time. As an example, Figure 2 shows the influent concentrations of the drugs carbamazepine (antiepileptic), diclofenac (anti-inflammatory), and estrone (estrogen) for the indicated sampling dates.

As can be seen, carbamazepine was quantified in all the samples, presenting two very prominent concentration peaks coinciding with the Easter holidays and the beginning of spring season. In the summer season, when the maximum tourist occupancy of Benidorm was reached, the concentration of the antiepileptic was five times lower, which indicates that the summer occupational group consumed much less of this medication. Regarding the anti-inflammatory diclofenac, three periods of time with high values were observed, the aforementioned spring period of April, the beginning of March, and the first week of May. It should be noted that in Spain there is a government aid program for retired people to promote holidays during periods of lower tourist occupancy, and Benidorm is one of the favorite destinations. Finally, estrone presents an irregular occurrence, with approximately half of the samples in non-quantifiable concentrations, and with three maximum concentrations that correspond to the week before those registered with diclofenac.

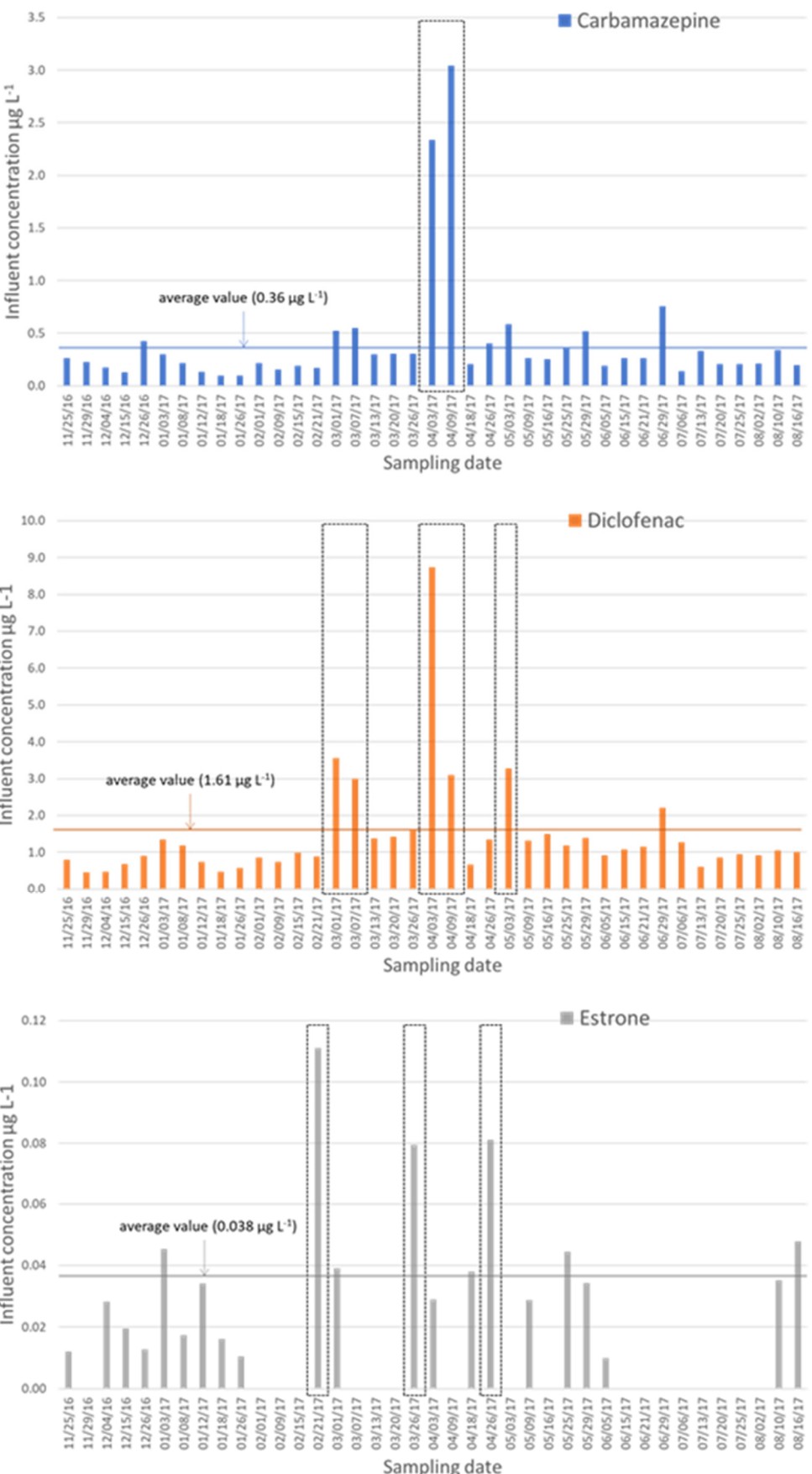

**Figure 2.** Influent concentrations of the drugs carbamazepine, diclofenac, and estrone for the indicated sampling dates.

## 4. Conclusions

The PSs atrazine, brominated diphenyl ether, isoproturon, octylphenol, pentachloro-benzene, simazine, terbutryn, tributyltin, and trifluralin, and the CECs 17-$\alpha$-ethinylestra-diol, 17-$\beta$-estradiol, imazalil, orthophenylphenol, tertbutylazine, and thiabendazole, were not detected at the Benidorm WWTP.

The chlorpyrifos was present in the effluent in concentrations exceeding the established environmental quality standards. Therefore, to reduce the concentration of this PS to environmentally acceptable values, it would be necessary to apply additional wastewater treatment to the effluent of the biological process. These additional treatments would also be necessary to reduce effluent concentrations of the CECs diclofenac, erythromycin (both included in the first EU observation list), sulfamethoxazole (included in the third EU observation list), carbamazepine, fluoxetine, ibuprofen and ketoprofen.

Some micropollutants were found in a high or very high concentration range compared to the values reported in the literature. This seems to be associated with the significant presence of elderly residents and tourists at certain times of the year.

**Author Contributions:** E.T.A.: Writing—original draft preparation. M.d.l.Á.B.-R.d.H.-B.: Investigation, data curation. R.G.: Funding acquisition, resources, project administration. A.V.S.: Resources, formal analysis. H.G.: Investigation, visualization writing—review and editing. D.P.: Conceptualization, supervision. All authors have read and agreed to the published version of the manuscript.

**Funding:** This research was funded by European Commission, grant number LIFE15ENV/ES/00598 Development of an Efficient and Sustainable Methodology for Emerging Pollutants Removal in WWTPS.

**Institutional Review Board Statement:** Not applicable.

**Informed Consent Statement:** Not applicable.

**Data Availability Statement:** 3$^{RD}$ Party Data. Data can be obtained from the European Unions EASME.

**Acknowledgments:** This research was assisted by Entidad Pública de Saneamiento de la Comunidad Valenciana, EPSAR. Benidorm City Council has provided all the sampling and has supplied the information with regards to the conditions of the operation of the plant.

**Conflicts of Interest:** The authors declare no conflict of interest.

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
