# Peer review of "Occurrence and Removal of Priority Substances and Contaminants of Emerging Concern at the WWTP of Benidorm (Spain)"

_water, doi:10.3390/w14244129_

Round 1
Reviewer 1 Report
The authors proposed the “Occurrence and removal of Priority Substances and Contaminants of Emerging Concern at the Benidorm (Spain) WWTP”. Overall, the whole content of the manuscript is interesting and the research scope is suitable to Water, and the results offer insight into development of wastewater treatment. Therefore, I think that minor revision is requested for the paper before accepted for publication. A few suggests and comments are listed as follows:
(1) Line 42, 220, the “.” need to be removed before the “()”.
(2) Line 59, you need to put the “.” after the “()”.
(3) In the references, you need to check punctuation for omissions (e.g. 21th, 22th, 25th).
(4) Please correct the citation format of reference 13.
(5) The quality of the Figures needs to be improved.
(6) When using analytical instruments, whether the sample needs to be pre-treated?
(7) The title needs to be refined.
Reviewer 2 Report
This is an interesting work focusing on the detection and quantification of CECs and PS from directives but some aspects should be revised.
The abstract should present more clear numerical results.
The introduction should be compared with other studies regarding to other regions of world.
The introduction the authors can refer the advantaged of tertiary treatment see these reviews to help on this discussion, Oller et al. 2011, Sci. Total Environ.409, 4141–4166; Gomes et al. 2017, Science of the total environment, 586 (2017) 265–283.
Why authors do not analysis the CECs and PS after the tertiary treatment. A comment should be added.
What the authors expected in terms of results of removal for primary treatment. It would be interesting understand if the primary removes some of CECs or PS.
Why do not compare the results of WWTP of Benidorm with other present in the literature. Should be added a section that highlights the comparison of this work results and previous founded in the literature.
The detection and quantification of detected product should be presented in the year season instead of medium along the year. During the year the concentrations can change since for example some pharmaceuticals are specific from the winter.
Reviewer 3 Report
The present study is based on a comprehensive data set covering one year of data for a broad spectrum of micropollutants. The study can be useful as a complement to previous studies on removal of organic micropollutants in WWTPs and probably be very useful as a national/local reference material. The scientific novelty as such is however low.
L. 63 – This statement is important and correct, but it would be good to elaborate on “type of treatment and operating conditions”. When can we expect higher removal? Lower removal? Why? A few lines on the importance of for example SRT would be in place.
L. 64 – L. 71 The objective of the LIFE-project “was to develop an efficient and sustainable methodology to eliminate PS and CEC in WWTPs.” Then it says “The aim was to achieve reductions in PS until their concentrations was below…” I assume the latter statement also refers to the overall project? Later in the same section it says that occurrence and removal of a number of substances are presented in this study. Was this, to present occurrence and removal, the aim of this study? I guess so, but it is maybe not entirely clear to the reader.
Figure 1 in the Material and Methods section is nice, but I miss a clear description of the secondary treatment at Benidorm WWTP somewhere in this chapter. What about retention times and loading conditions? It says that it is a CAS-plant, but that is a rather vague description. Expected removal can vary a lot depending upon type of AS-process and actual operating conditions, just as indicated in the introduction.
Table 2 is nice and clear. Could you please comment on the relatively high LOQs for some substances, e.g. Carbamazepine?
Table 3. Removal of CBZ is not expected, but was the influent and effluent average concentrations, and standard deviations, really identical? Erythromycin also looks very similar. This can of course be the case but I ask just to make sure influent or effluent values haven’t been duplicated.
L.148 and on – it would be good for the reader with a somewhat more elaborated discussion on some of the substances with regards to operational parameters. See above.
L.264 and on – This part is interesting and deserves a separate headline. What about effluent concentrations (removal) during the peaks? Could you provide a short discussion and how this knowledge can be used when for example applied in design (as indicated in the introduction)?
L. 302 - 309 These sections are vague and more of a repetition of results rather than conclusions.
Round 2
Reviewer 3 Report
The paper has improved following the review.
Your response
Some of the operating parameters in a real treatment plants, such as Benidorm, fluctuate over time, for example flows, organic loads and temperatures. The daily values can vary a lot depending on the season of the year or the periods of greater or lesser occupation. Also the concentrations of micropollutants are very variable, reaching very different values in the different samples. It is for this reason that the article presents the average values, which are more useful than the point values for the purpose of seeking an effective tertiary treatment. Consequently, it would be very complex to try to relate real operating conditions (very fluctuating) and concentrations of micropollutants (also very fluctuating) with the removals obtained (which also vary in the different samplings). A more elaborate discussion would be possible under experimental conditions programmed in a pilot plant, which is not our case.
is obviously correct (although it would be possible to pinpoint some interesting observations in your extensive data set) but it should not stop you from at least indicating the SRT practised at the plant. From the description added it seems as activated sludge process is based on nitrification/dentrification for nitrogen removal. If so, this is important to stress. It is probably a low-loaded activated sludge process allowing for nitrification (?) which has a positive effect on removal of certain micropollutants. In the method description an indication of sludge age is in place.
